# Non-Covalent Interactions in the Crystal Structures of Perbrominated Sulfonium Derivatives of the *closo*-Decaborate Anion

**DOI:** 10.3390/ijms231912022

**Published:** 2022-10-10

**Authors:** Aleksei V. Golubev, Alexey S. Kubasov, Alexander Yu. Bykov, Andrey P. Zhdanov, Grigorii A. Buzanov, Alexander A. Korlyukov, Konstantin Yu. Zhizhin, Nikolay T. Kuznetsov

**Affiliations:** 1Kurnakov Institute of General and Inorganic Chemistry, Russian Academy of Sciences, 31 Leninsky Prosp., 119071 Moscow, Russia; 2A.N. Nesmeyanov Institute of Organoelement Compounds, Russian Academy of Sciences, 28 Vavilova Str., 119991 Moscow, Russia

**Keywords:** boron cluster, *closo*-decaborate anion, perbrominated derivatives, X-ray, DFT calculations

## Abstract

A new series of compounds based on perbrominated disubstituted sulfonium derivatives of the *closo*-decaborate anion (*n*-Bu_4_N)[2-B_10_Br_9_SR_2_] (R = *n*-Pr, *i*-Pr, *n*-Bu, *n*-C_8_H_17_, *n*-C_12_H_25_, *n*-C_18_H_37_) was obtained, characterised by modern physicochemical methods of analysis. According to the results of an X-ray diffraction study, some of the anions and solvate molecules were disordered. The cations (*n*-Bu_4_N)^+^ and anions [2-B_10_Br_9_SR_2_]^−^ were associated via C-H…Br and H…H contacts. In addition, Br…Br interactions between anions were revealed. The role of these contacts was analysed in terms of Hirshfeld surface analysis, QTAIM theory and the NCI method using quantum chemical calculations. An increase in the size of the alkyl R moiety led to significant strengthening of the total energy of H…H interactions. In the case of R = -*n*-C_18_H_37_, a parallel mutual orientation of alkyl moieties was established that was similar to the packing of salts of fatty acids. The nature of C-H…Br and Br…Br interionic interactions was found to be attractive, in contrast to the repulsive nature of intermolecular Br…Br interactions.

## 1. Introduction

Non-covalent interactions continue to receive significant attention, due to their crucial role in various fields of chemistry, such as the stabilisation of metal complexes with various types of ligands, or the interaction of halogens with Lewis acids, which have applications in crystal engineering [1,2], organic synthesis [3] and chemistry materials [4].

An example of a context where non-covalent interactions play an important role is provided by compounds that are based on boron cluster anions. In this case, various types of weak interactions are formed, depending on the type of compound used [5,6,7,8]. Due to the large number of possible modifications by introducing exo-polyhedral substituents [9,10,11], a wide range of possible applications have been found for these compounds. They can be used to obtain ionic liquid crystals [12,13,14], as a component in the creation of membranes for the highly sensitive detection of drugs [15] or for the creation of liquid crystalline NLO chromophores [16,17]. In addition, the complete replacement of hydrogen atoms in the boron cage leads to a significant decrease in its coordinating ability, as a result of which they can be used to stabilize reactive cations [18,19,20]. Combining these methods makes it possible to obtain a new type of compound, which, for example, can be used to obtain ionic liquids and ionic crystals [21,22,23].

Most of the works in this area are devoted to the study of perchlorinated derivatives of *closo*-borate anions, and only a few to perbrominated anions. The most extensive studies of the properties of this class of compounds are devoted to the study of perhalogenated *alkoxy*- [24] and ammonium [25,26] derivatives of *closo*-borate anions. However, there are very few theoretical studies in this area for these anions. Our work is intended to bring some clarity into the nature of X…X, X…H interactions for such compounds. The study of the influence of non-covalent interactions on molecular structuring in these materials will expand the fundamental concepts of the use of compounds based on boron cluster anions, and will enable the future creation of new types of materials with desired physicochemical properties.

Previously, the authors have studied perchlorinated sulfonium derivatives of the *closo*-decaborate anion with alkyl substituents with different hydrocarbon chain lengths [27]. The current article presents the synthesis and study of their perbrominated analogues. Analysis of the compounds obtained using X-ray diffraction analysis, Hirschfeld surface analysis and quantum chemical calculations will make it possible to establish the role of non-covalent interactions in the molecular assembly of these compounds and enable an understanding of the mutual arrangement of molecular fragments, which is of fundamental importance for the development of materials based on compounds of this type.

## 2. Results

**Synthesis.** The process of complete bromination of tetrabutylammonium salts of sulfonium derivatives of the *closo*-decaborate anion (*n*-Bu_4_N)[2-B_10_H_9_SR_2_] (R = *n*-Pr, *i*-Pr, *n*-Bu, *n*-C_8_H_17_, *n*-C_12_H_25_, *n*-C_18_H_37_) is most conveniently carried out using elemental bromine in acetonitrile (Figure 1). Before starting to add a solution of elemental bromine in acetonitrile, it is necessary to cool the reaction solution to 0 °C in an ice bath, since the start of the reaction is accompanied by the release of a large amount of heat and gas (H_2_, Br_2_, HBr). After complete pouring of the reagent solution (a solution of elemental bromine in acetonitrile) to the mixture, the solution is slowly warmed to room temperature and after 24 h the only final reaction product is a completely substituted derivative.

The ^11^B NMR spectra of the final compounds practically did not differ in their patterns from their perchlorinated counterparts (see Appendix A). The spectrum shows two signals at −0.5 and −4.9 ppm, which refer to the apical peaks. The signals from boron atoms in the equatorial belt are at −10.6, −13.2 and −17.0 ppm and do not change in their interpretation from the polyhedron, which is B4, the group of atoms B3, B5, B6–B9, and the ipso-atom B2, respectively.

Compared to the perchlorinated sulfonium derivatives of the *closo*-decaborate anion, the following pattern was observed in the ^1^H NMR spectrum of the final compounds after the bromination process (see Appendix A): the signal from the protons of the α-methylene groups shifted to the downfield region by 0.2 ppm (compared to the non-halogenated derivatives, by 0.9 ppm). In addition, in perbrominated derivatives, a change in the signal from the protons of β-methylene groups was observed, which was expressed in the formation of two complex multiplets at 1.94 and 1.87 ppm, which was not observed either in the initial sulfanyl-*closo*-borates, or in their perchlorinated counterparts. The other signals remained unchanged.

In ^13^C NMR spectra of perbrominated derivatives, as compared to perchlorinated salts, the signals from alpha-methylene groups were shifted by approximately 0.1–0.2 ppm (see Appendix A).

In the IR spectra of the final compounds, after exhaustive bromination processes, the disappearance of absorption bands in the region of 2600–2400 cm^−1^, characteristic of the ν_B-H_ stretching vibrations, and the formation of three strong bands at 1123, 970 and 433 cm^−1^ were observed in the example of the compound **1** characteristic of ν_B-Br_ vibrations (see Appendix A).

**Melting point measurements.** For all obtained salts of perbrominated disubstituted sulfonium derivatives of the *closo*-decaborate anion with tetrabutylammonium cation (*n*-Bu_4_N)[2-B_10_Br_9_SR_2_] (R = *n*-Pr, *i*-Pr, *n*-Bu, *n*-C_8_H_17_, *n*-C_12_H_25_, *n*-C_18_H_37_), melting points were examined and found to be virtually identical to their perchlorinated analogues. Compounds with short hydrocarbon chains (up to *n*-Bu) had a melting point above 200 °C, while for a compound with –*n*-C_8_H_17_ substituents it was 127–128 °C; for –*n*-C_12_H_25_ it was 98–99 °C and for –*n*-C_18_H_37_ it was 60–62 °C.

**X-ray diffraction analysis and Hirschfeld surface analysis.** Structure of anions (*n*-Bu_4_N)[2–B_10_Br_9_S(*n*-Pr)_2_] (**1**), (*n*-Bu_4_N)[2–B_10_Br_9_S(*i*-Pr)_2_] (**2**), (*n*-Bu_4_N)[2–B_10_Br_9_S(*n*-Bu)_2_] (**3**), (*n*-Bu_4_N)[2–B_10_Br_9_S(*n*-C_12_H_25_)_2_] (**5**), and (*n*-Bu_4_N)[2–B_10_Br_9_S(*n*-C_18_H_37_)_2_] (**6**) are shown in Figure 1.

The crystallographically independent parts of the monoclinic unit cell (P2_1_/c) salt **5** and triclinic cells (P-1) of salts **1**, **2**, **3**, and **6** contain one cation and one anion each. The cell of the last salt also includes a highly disordered DMSO molecule that cannot be modelled. Hence, an integral contribution of the strongly disordered DMSO molecule was excluded from total electron density; it was removed from the electron density map using the solvent masking implemented in the OLEX2 program. Exopolyhedral iso-propyl groups of the [2–B_10_Br_9_S(*i*-Pr)_2_]^−^ anion were disordered and rotated relative to one another, so that the C1C2C3 and C4C5C6 planes formed angles of 61.2 and 66.0°.

In all of the compounds obtained, the geometry of the boron core was not distorted; the lengths of the B–B bonds corresponded to the unsubstituted [B_10_H_10_]^2−^ anion and derivatives [2-B_10_H_9_SR_2_]^−^ (1.656–1.704 for the bonds of the apical apex and 1.802–1.865 for the bonds of the equatorial belt (Table 1)). The B-Br bond lengths for the apical boron atoms B1 and B10 lay in the range of 1.925–1.956 Å, which was, on average, slightly shorter than the corresponding bonds for the equatorial boron atoms B3-B9, which lay in the range of 1.933–1.969 Å, which was consistent with other perbrominated derivatives of the sulfonium-type *closo*-decaborate anion [26,28,29]. The B-S bond lengths lay in the range of 1.887–1.912 Å, which also corresponded to analogous bonds in other derivatives of the sulfonium-type *closo*-decaborate anion [27,28]. The S–C bond lengths lay in the range of 1.814–1.838 Å.

The structural motif of the salt **1** (Figure 2a) is similar to its perchlorinated analogue (*n*-Bu_4_N)[2–B_10_Cl_9_S(*n*-Pr)_2_] [27] and compounds **2** (Figure 3a) and **3** (Figure 4a). In these cases, the anion was formed by dimer pairs bound by weak CH…Hal contacts and linked to one another by analogous contacts with (*n*-Bu_4_N)^+^ cations and Hal…Hal contacts, (the length of Br…Br contacts in compound **1** was 3.55 and 3.71 Å; it was 3.62 and 3.94 Å in compound **2**, and 3.59 and 3.69 Å in compound **3**). The Br…Br contacts are shown on the Hirschfeld surfaces of the anions in Figure 2b, Figure 3b and Figure 4b. Analysis of the fingerprint plots shows that Br…Br contacts accounted for 4.4%, 2.1% and 3.7% of all contacts in compound **1**, **2** and **3**, respectively. The Br…H/H…Br contacts accounted for the largest percentage of the anion surface, at 67.9, 72.5 and 67.5%, respectively.

In compounds **5** and **6**, layers were formed. One of these was built from exopolyhedral alkyl substituents, while the second layer consists of [2-B_10_Br_9_SR_2_]^−^ anions and (*n*-Bu_4_N)^+^ cations alternating in a checkerboard pattern. If, in the first case, the alkyl groups “shielded” the layers from each other, then in the second case their interweaving is observed as part of the supramolecular bilayer. A further feature in the structures was that, in the packing of compound **6**, the centres of cations and the boron cores were located approximately at the same level in the bilayer (Figure 5). In compound **5**, the cation centres were located on the plane passing through the centre of the unit cell, parallel to the BC plane, and the boron cores were located on the planes at approximately 1/3 and 2/3 of the way along the a axis.

A detailed analysis of the Hirschfeld surface of the anion showed that, just as in the cases described above, the boron cores were connected by weak CH…Br and Br…Br contacts between themselves and tetrabutylammonium cations (Figure 6). It is obvious that, with an increase in the length of the alkyl substituent in the structures, the proportion of H…H contacts increased (Figure 7).

**Experimental vs. optimised structures.** The structures were ionic, and some were partly disordered, so the analysis of differences between the calculated and experimental structure was not a straightforward procedure using R.M.S. As a result of cell vector relaxation, calculated cell volume appeared to be 5–7% less than the experimental figure. The reason for this was shortcomings in the DFT functional/D3 dispersion correction and temperature contraction, (in fact, the calculations were carried out at 0 K). Despite this, the PW-PBE-D3 calculations enabled the accurate reproduction of most of the bond lengths and angles in the cationic and anionic fragments of the structures studied. Most of the bonds in both anionic and cationic fragments were reproduced to within 0.03 Å. The values corresponding to short anion–anion and cation–anion interactions appeared to be smaller than in the experimental structures. For instance, the experimentally derived interatomic distances corresponding to the C-H…Br interactions between cations and anions were 0.2–0.3 Å larger than in the calculated structure. The deviations in Br…Br distances between the experimentally derived figures and the calculated figures varied in the range 0.03–0.28 Å. Thus, PW-PBE calculations can somewhat overestimate the energy of cation-anion interactions that can be explained by removing of disorder and cell contraction. Nevertheless, the accuracy is sufficient to describe the features of crystal packing in compounds studied.

**Study of the bonding between charged molecular fragments: QTAIM and NCI study.** Quantum theory “Atoms in molecules” (QTAIM) [30] has been proven to be a reliable tool for revealing and describing the characteristics of most types of interatomic interaction. This type of study is complementary to Hirshfeld surface analysis, which enables a more detailed analysis of the nature of intermolecular interaction. The localisation and description of interatomic interaction relies on an analysis of the gradient of the electron density function (∇ρ(r)) to reveal the regions of maximum gradient (gradient paths) connecting two atoms. The point with a zero value of ∇ρ(r) on the gradient path (critical point (3,−1) or bond critical point (bcp)) is an indicator of a chemical bond or a weak interatomic interaction. The strength of the chemical bond or interatomic interaction is defined by the value of ρ(r) in bcp or related kinetic or potential energies. It is very useful to rationalise the role of weak H…A (A = O, N, Hal), H…H and Hal…Hal interactions in crystal packing. An estimation of the energy of such weak interactions is possible using two empirical formulas derived from the correlation between kinetic/potential energy densities in bcp and interatomic distance [31,32].

In terms of the QTAIM approach, the electron density inside the atomic domain can be integrated, thus giving the value of the atomic charge. In a similar way to previously studied compounds with B_10_Cl_9_ fragments [33], QTAIM charges were calculated for all atoms. Due to the presence of an electron-deficient boron core, the charge of bromine atoms varied from positive (up to +0.21 e) to negative (−0.34 e), depending on the position of the corresponding boron atom. The most negatively charged bromine atoms were found in the vicinity of a sulphur atom. Most of the boron atoms were positively charged, but two of them had a small negative charge (up to −0.11 e). These atoms are located on the opposite side to the sulphur atom in boron cluster.

Negatively charged brominated polyhedral boron fragments and positively charged alkyl fragments were responsible for cation–anion and anion–anion interactions in the crystals of salts studied.

The energies of separate interatomic interactions estimated from the QTAIM study can be used to calculate the interactions between charged molecular fragments (cations and anions). These calculated energy values are presented in Figure 8. As was expected, the increase in the size of the alkyl chain at the sulphur atom led to a gradual increase in the interaction energy. The increase in the energy of anion–anion interaction was more pronounced than in other energies. The reason is that the number of H…H interactions between alkyl chains significantly increases upon going from (*n*-Bu_4_N)[2–B_10_Br_9_S(*i*-Pr)_2_] to (*n*-Bu_4_N)[2–B_10_Br_9_S(*n*-C_18_H_37_)_2_].

In addition to H…H interactions, Br…H and Br…Br interactions were responsible for anion–anion and cation–anion bonding. Most Br…H interactions are weak (less than 5 kJ/mol), and they are mostly responsible for cation–anion bonding. Conversely, Br…Br interactions play a noticeable role in the association between anions. In most cases, the energy of Br…Br does not exceed 5 kJ/mol, but in salts studied in the current work there were several interactions that were considerably stronger. For example, the energy of the Br6…Br6 interaction in (*n*-Bu_4_N)[2–B_10_Br_9_S(*n*-C_12_H_25_)_2_] was 9.8 kJ/mol. In addition to intermolecular interactions, some intramolecular H…H and Br…H interactions were revealed. The energy of the strongest interactions of this type was up to 2.4 kcal/mol, which exceeded the values for analogous intermolecular interactions.

The presence of sufficiently bulky alkyl substituents near sulphur atoms led to the formation of numerous intermolecular interactions that were both intermolecular and intramolecular. Most of these were of the Br…H type. In addition, in the case of -C_18_H_37_, the H…H interactions between hydrogen atoms of alkyl chains led to their parallel mutual orientation. The total energy of such interactions between adjacent alkyl chains of anions [2–B_10_Br_9_S(*n*-C_18_H_37_)_2_] was 80 kJ/mol.

To visualise and describe the nature of the interactions in a comprehensive manner, the features of electron density distribution were analysed using the NCI method [34]. Similar to QTAIM, the NCI method relies on a calculated, or experimental, electron density distribution function. The use of the gradient of ∇ρ(r) enables an emphasis on the change in the distribution of ρ(r) upon the formation of an intermolecular interaction using empirical scaling, thus obtaining a reduced density gradient (RDG) function. To distinguish the nature of intermolecular interactions (repulsive or attractive), the isosurfaces created using an RDG function are usually coloured according to the values of the sign(λ_2_) ρ(r) function. The latter is the product of multiplications of the values of λ_2_ eigenvalues of the Hessian matrix (it can be positive or negative) to the values of ρ(r) at a given point. With the NCI method, any non-covalent interaction is defined as the region between atoms, in contrast to the QTAIM definition, where bcp is indicative of interaction. Attractive interactions are characterised by negative values of λ_2_, while repulsive interactions are described by positive λ_2_.

Thus, the analysis of interatomic interactions using the NCI method indicated the attractive character of Br…Br interactions between borane moieties. One of the strongest Br…Br interactions is shown in (Figure 9). It can be seen that the isosurface of the RDG function has the form of a double convex lens, coloured in blue to indicate that λ_2_ < 0. The isosurfaces of the RDG function related to intermolecular Br…H interactions can be described as being ‘attractive’. In addition, the NCI study revealed intermolecular Br…Br interactions between adjacent bromine atoms that can be described as being mostly repulsive (λ_2_ > 0).

## 3. Discussion

In contrast to chlorination of boron cluster anions [B_n_H_n_]^2-^ (n = 10, 12) with elemental chlorine, which requires a significant excess of the reagent and a temperature above 100 °C [35], bromination proceeds from sulfonium derivatives of *closo*-decaborate anions [2-B_10_H_9_SR_2_]^−^ at a threefold excess of bromine at room temperature for 24 h. If a smaller amount of elemental bromine is used, a mixture of fully and partially halogenated substituted derivatives is observed in the reaction mixture over the same time period. With an elemental bromine:boron cluster ratio of 2:1, the halogenation process can also be completed, but it is necessary to increase the reaction time to 3 days or more.

Investigation of the melting points of perchlorinated [27] and perbrominated alkylsulfonium derivatives of the *closo*-decaborate anion has shown that they decrease significantly with an increase in the length of the alkyl substituent. This trend is also confirmed for trisubstituted ammonium salts [B_12_H_11_NR_3_]^−^ [21,36] and salts of alkoxy derivatives of the *closo*-dodecaborate anion [B_12_Cl_11_OR]^−^ and [B_12_Br_11_OR]^−^ [24]. This makes it possible to classify this class of compounds, even in the case of tetrabutylammonium salts, as ionic liquids. When using more suitable cations, such as immidazolium cations, melting points can be significantly reduced.

At first glance, NCI analysis of intermolecular interactions in the obtained compounds shows that the values of the structural properties of anions (surface area, volume, total energy of intermolecular interactions) increase, but from a thermodynamic point of view, it seems that the melting temperature values are determined by the molecular flexibility of alkyl substituents in anions, which can be described as conformational entropy.

## 4. Materials and Methods

**(*n*-Bu_4_N)[2-B_10_H_9_SR_2_] (R = *n*-Pr, *i*-Pr, *n*-Bu, *n*-C_8_H_17_, *n*-C_12_H_25_, *n*-C_18_H_37_)** was obtained by a known method [37]. Bromine (99.5%) was purchased from Sigma-Aldrich (St. Louis, MO, USA) and used as received. Acetonitrile was boiled under reflux over calcium hydride to remove residual water for several hours and distilled at atmospheric pressure. All other reagents and solvents were commercially available and used without further purification.

**Elemental analysis** for carbon, hydrogen, nitrogen and sulphur was carried out on a Carlo Erba CHNS3 FA 1108 Elemental Analyser.

**IR spectra** were recorded on an Infralyum FT 02 Fourier transform spectrometer (Lumex Instruments Research and Production Company, St. Petersburg, Russia) in the range 4000–400 cm^–1^ at a resolution of 1 cm^–1^. Samples were prepared as suspensions in CCl_4_.

**^1^****H, ^11^B and ^13^C NMR spectra** of samples dissolved in DMF-d7 were recorded on a Bruker Avance II 300 spectrometer operating at a frequency of 300.3, 96.32 and 75.49 MHz, using an internal deuterium lock. Tetramethylsilane and boron trifluoride etherate were used as external references. The spectra of the compounds obtained are presented in Appendix A.

**Melting point** was measured on a REACH Devices RD-MP in the temperature range of 25−250 °C with a heating interval of 1.5, 3, 6 and 12 °C/min.

**The X-ray diffraction.** Single crystals were obtained by crystallisation from dimethylsulfoxide solution. X-ray diffraction data were collected with a Bruker SMART APEX II diffractometer using graphite monochromated Mo Kα radiation with multilayer optics. The structures were solved using the direct method and refined using the full-matrix least squares method against F^2^ of all data, using SHELXL-2014 [38] and OLEX2 [39] software. Non-hydrogen atoms were found on difference Fourier maps and refined with anisotropic displacement parameters. The positions of hydrogen atoms were calculated and included in refinement in isotropic approximation by the riding model, with Uiso(H) = 1.5 Ueq(C) for methyl groups and 1.2 Ueq(Cii) for other atoms, where Ueq(C) are equivalent thermal parameters of parent atoms. 

One molecule of DMSO in the asymmetric unit of (*n*-Bu_4_N)[2–B_10_Br_9_S(*n*-C_18_H_37_)_2_] was found to be disordered over multiple positions and could therefore not be modelled satisfactorily. It was removed from the electron density map using the OLEX solvent mask command. 

Details of data collection and refinement are listed in Appendix A. The crystallographic data for X were deposited with the Cambridge Crystallographic Data Centre as supplementary publications under the number CCDC 2166814-2166818.

**The X-ray phase analysis.** Analytical research (pXRD) was done using equipment from the NRC “Kurchatov Institute”—IREA Shared Knowledge Centre. Using a Bruker D8 ADVANCE X-ray diffractometer (CuKalpha radiation, Ni-filter, reflection geometry, LYNXEYE detector, 298 K) in the angle range 2Theta = 5–50 deg., with a step of 0.01125 deg. and accumulation time 0.1–0.25 s. in cuvettes with a substrate of oriented single-crystal silicon. Immediately before recording, the test samples were ground in a synthetic jasper mortar and placed in cuvettes, levelling the sample level flush. Diffractograms of the compounds obtained are presented in Appendix A.

**Hirshfeld surface analysis.** The Crystal Explorer 17.5 program [40] was used to analyse the interactions realized within the crystal. The donor-acceptor groups were visualised using a standard (high) surface resolution, and d_norm_ surfaces were mapped over a fixed colour scale of –0.640 (red) to 0.986 (blue) a.u.

**Computational details.** To describe the features of crystal packing, four crystal structures were selected: (*n*-Bu_4_N)[2–B_10_Br_9_S(*i*-Pr)_2_] (2), (*n*-Bu_4_N)[2–B_10_Br_9_S(*n*-Bu)_2_] (3), (*n*-Bu_4_N)[2–B_10_Br_9_S(*n*-C_12_H_25_)_2_] (5) and (*n*-Bu_4_N)[2–B_10_Br_9_S(*n*-C_18_H_37_)_2_] (6), which have alkyl substituents of different lengths. Between (*n*-Bu_4_N)[2–B_10_Br_9_S(*i*-Pr)_2_] and (*n*-Bu_4_N)[2–B_10_Br_9_S(*n*-Pr)_2_], it was decided to calculate only the first one to save computational time, because *i*-Pr and borane form intramolecular interactions that can be characterised through a study of the electron density function. All quantum chemical calculations were performed at 0 K.

The structures studied with quantum chemical calculations were characterised by substantial uncertainties relating to atomic coordinates, due to thermal motion and because it was impossible to precisely refine the position of hydrogen atoms. Some structures were disordered. To overcome these problems, the disorder of some molecular fragments was removed and the atomic coordinates were optimised. After that, all atomic coordinates and cell vectors were relaxed. The values of optimised atomic coordinates, cell vectors and total energies can be found in Appendix A.

Projector augmented wave (PAW) pseudopotentials [41,42] were used for all atoms to describe core electrons. The contribution of valence electrons was described as a series of plane waves with kinetic energy cut-off of 800 eV. Exchange and correlation terms of total energy were described by PBE functional [43], with Grimme D3 dispersion correction [44]. Electron density function for topological analysis was obtained using a separate single point calculation with tighter convergence criteria (10^−7^ eV). DFT (PW-PBE-D3) periodic calculations were performed using the VASP package [45,46,47,48]. Topological analysis of calculated electron density function was carried out using the AIM program (part of ABINIT software) [49]. 

**Synthesis. Safety Notes.** Caution: Carrying out the bromination reaction using bromine as a halogenating agent in the first stages proceeds with a violent evolution of heat and gas; it is necessary to cool the reaction solution and add Br_2_ at a slow rate.


**General method for the synthesis of compounds (*n*-Bu_4_N)[2-B_10_Br_9_SR_2_].**


Tetrabutylammonium salt with the appropriate alkyl substituents was placed in a 25 mL round bottom flask and dissolved in 3 mL of acetonitrile. The resulting solution was cooled to 0 °C in an ice bath. A solution of elemental bromine in acetonitrile was added dropwise to the reaction mixture in an argon atmosphere. The reaction mixture was then slowly warmed to room temperature and allowed to stir for 24 h. Then, the solution was evaporated until the complete removal of volatile reaction products. The resulting solid residue had 10 mL of hexane added to it, before it was placed in an ultrasonic bath for 10 min. The hexane fraction was then decanted and the procedure was repeated once more. Then, 10 mL of distilled water and 10 mL of diethyl ether were added to the solid residue, and it was left in an ultrasonic bath for a further 10 min. The precipitate formed was filtered off and washed several times with, successively, 2 × 10 mL of hexane, 2 × 10 mL of distilled water and 2 × 10 mL of diethyl ether.

**(*n*-Bu_4_N)[2-B_10_Br_9_S(*n*-Pr)_2_] (1).** From (*n*-Bu_4_N)[2-B_10_H_9_S(*n*-Pr)_2_] (200 mg, 0.42 mmol) and Br_2_ (568 µL, 11.34 mmol), (*n*-Bu_4_N)[2-B_10_Br_9_S(*n*-Pr)_2_] (427 mg, 0.36 mmol) was obtained. Yield: 85%. Anal. calc. of C22H50B10Br9NS: C, 22.2; H, 4.2; N, 1.2; S, 2.7; Found: C, 22.0; H, 4.1; N, 1.0; S, 2.5. ^11^B NMR (DMF-d7, 96.32 MHz) ppm: −0.5 (s, 1B), −4.9 (s, 1B), −10.6 (s, 1B), −13.2 (s, 6B), −17.0 (s, 1B). ^1^H NMR (DMF-d7, 300.3 MHz) ppm: 3.56 (m, 2H, SCH_2_), 3.56 (m, 2H, SCH_2_), 3.40 (m, 8H, Bu_4_N^+^), 1.94, 1.87 (m, 4H, SCH_2_CH_2_), 1.77 (m, 8H, Bu_4_N^+^), 1.39 (m, 8H, Bu_4_N^+^), 1.04 (t, 6H, CH_3_), 0.97 (t, 12H, Bu_4_N^+^). ^13^C NMR (DMF-d7, 75.49 MHz) ppm: 59.3 (Bu_4_N^+^), 41.0 (SCH_2_), 24.6 (Bu_4_N^+^), 21.7 (SCH_2_CH_2_), 20.6 (Bu_4_N^+^), 14.2 (Bu_4_N^+^), 13.4 (CH_3_). IR (CCl_4_) cm^−1^: 2961, 2931, 2873, 1470, 1378, 1308, 1242, 1155, 1123, 1067, 970, 880, 795, 759, 433.

**(*n*-Bu_4_N)[2-B_10_Br_9_S(*i*-Pr)_2_] (2).** From (*n*-Bu_4_N)[2-B_10_H_9_S(*i*-Pr)_2_] (200 mg, 0.42 mmol) and Br_2_ (568 µL, 11.34 mmol), (*n*-Bu_4_N)[2-B_10_Br_9_S(*i*-Pr)_2_] (444 mg, 0.37 mmol) was obtained. Yield: 89%. Anal. calc. of C22H50B10Br9NS: C, 22.2; H, 4.2; N, 1.2; S, 2.7; Found: C, 22.0; H, 4.0; N, 1.0; S, 2.5. ^11^B NMR (DMF-d7, 96.32 MHz) ppm: −0.5 (s, 1B), −5.9 (s, 1B), −10.2 (s, 1B), −13.2 (s, 6B), −17.2 (s, 1B). ^1^H NMR (DMF-d7, 300.3 MHz) ppm: 4.32 (m, 4H, SCH), 3.40 (m, 8H, Bu_4_N^+^), 1.81 (d, 12H, CH_3_), 1.78 (m, 8H, Bu_4_N^+^), 1.39 (m, 8H, Bu_4_N^+^), 0.97 (t, 12H, Bu_4_N^+^). ^13^C NMR (DMF-d7, 75.49 MHz) ppm: 59.3 (Bu_4_N^+^), 45.1 (SCH), 24.6 (Bu_4_N^+^), 23.2 (CH_3_), 20.6 (Bu_4_N^+^), 14.2 (Bu_4_N^+^). IR (CCl_4_) cm^−1^: 2961, 2932, 2873, 1469, 1378, 1307, 1240, 1156, 1120, 1066, 969, 896, 880, 797, 760, 432.

**(*n*-Bu_4_N)[2-B_10_Br_9_S(*n*-Bu)_2_] (3).** From (*n*-Bu_4_N)[2-B_10_H_9_S(*n*-Bu)_2_] (200 mg, 0.39 mmol) and Br_2_ (527 µL, 10.53 mmol), (*n*-Bu_4_N)[2-B_10_Br_9_S(*n*-Bu)_2_] (412 mg, 0.34 mmol) was obtained. Yield: 87%. Anal. calc. of C24H54B10Br9NS: C, 23.7; H, 4.5; N, 1.1; S, 2.6; Found: C, 23.6; H, 4.2; N, 1.0; S, 2.4. ^11^B NMR (DMF-d7, 96.32 MHz) ppm: −0.5 (s, 1B), −4.9 (s, 1B), −10.6 (s, 1B), −13.2 (s, 6B), −17.0 (s, 1B). ^1^H NMR (DMF-d7, 300.3 MHz) ppm: 3.58 (m, 4H, SCH_2_), 3.39 (m, 8H, Bu_4_N^+^), 1.92 (m, 2H, SCH_2_CH_2(a)_), 1.86 (m, 2H, SCH_2_CH_2(b)_), 1.77 (m, 8H, Bu_4_N^+^), 1.47 (m, 4H, CH_2_CH_3_), 1.42 (m, 8H, Bu_4_N^+^), 0.97 (t, 12H, Bu_4_N^+^), 0.95 (t, 6H, CH_3_),. ^13^C NMR (DMF-d7, 75.49 MHz) ppm: 59.4 (Bu_4_N^+^), 38.9 (SCH_2_), 30.0 (SCH_2_CH_2_), 24.6 (Bu_4_N^+^), 22.4 (CH_2_CH_3_), 20.6 (Bu_4_N^+^), 14.2 (Bu_4_N^+^), 13.9 (CH_3_). IR (CCl_4_) cm^−1^: 2959, 2932, 2872, 1470, 1417, 1380, 1340, 1309, 1242, 1229, 1150, 1122, 1032, 969, 896, 882, 795, 760, 435.

**(*n*-Bu_4_N)[2-B_10_Br_9_S(*n*-C_8_H_17_)_2_] (4).** From (*n*-Bu_4_N)[2-B_10_H_9_S(*n*-C_8_H_17_)_2_] (200 mg, 0.32 mmol) and Br_2_ (438 µL, 8.74 mmol), (*n*-Bu_4_N)[2-B_10_Br_9_S(*n*-C_8_H_17_)_2_] (365 mg, 0.27 mmol) was obtained. Yield: 86%. Anal. calc. of C32H70B10Br9NS: C, 28.9; H, 5.3; N, 1.0; S, 2.4; Found: C, 28.7; H, 5.1; N, 0.9; S, 2.2. ^11^B NMR (DMF-d7, 96.32 MHz) ppm: −0.4 (s, 1B), −4.9 (s, 1B), −13.2 (s, 8B). ^1^H NMR (DMF-d7, 300.3 MHz) ppm: 3.59 (m, 4H, SCH_2_), 3.39 (m, 8H, Bu_4_N^+^), 1.94 (m, 2H, SCH_2_CH_2(a)_), 1.87 (m, 2H, SCH_2_CH_2(b)_), 1.77 (m, 8H, Bu_4_N^+^), 1.42 (m, 8H, Bu_4_N^+^), 1.28 (m, 10H, C3-C7), 0.98 (t, 12H, Bu_4_N^+^), 0.88 (t, 6H, CH_3_). ^13^C NMR (DMF-d7, 75.49 MHz) ppm: 59.4 (Bu_4_N^+^), 39.2 (SCH_2_), 32.6 (SCH_2_CH_2_), 29.8, 29.5, 29.1, 28.0 (C3-C6), 24.7 (Bu_4_N^+^), 23.5 (CH_2_CH_3_), 20.6 (Bu_4_N^+^), 14.7 (CH_3_), 14.2 (Bu_4_N^+^). IR (CCl_4_) cm^−1^: 2958, 2932, 2874, 1470, 1417, 1380, 1362, 1340, 1309, 1242, 1122, 969, 896, 882, 795, 762, 435.

**(*n*-Bu_4_N)[2-B_10_Br_9_S(*n*-C_12_H_25_)_2_] (5).** From (*n*-Bu_4_N)[2-B_10_H_9_S(*n*-C_12_H_25_)_2_] (200 mg, 0.27 mmol) and Br_2_ (365 µL, 7.29 mmol), (*n*-Bu_4_N)[2-B_10_Br_9_S(*n*-C_12_H_25_)_2_] (319 mg, 0.22 mmol) was obtained. Yield: 82%. Anal. calc. of C40H86B10Br9NS: C, 33.3; H, 6.0; N, 1.0; S, 2.2; Found: C, 33.1; H, 5.8; N, 0.8; S, 2.0. ^11^B NMR (DMF-d7, 96.32 MHz) ppm: −0.4 (s, 1B), −4.9 (s, 1B), −13.2 (s, 8B). ^1^H NMR (DMF-d7, 300.3 MHz) ppm: 3.59 (m, 4H, SCH_2_), 3.39 (m, 8H, Bu_4_N^+^), 1.95 (m, 2H, SCH_2_CH_2(a)_), 1.88 (m, 2H, SCH_2_CH_2(b)_), 1.77 (m, 8H, Bu_4_N^+^), 1.39 (m, 8H, Bu_4_N^+^), 1.28 (m, 18H, C3-C11), 0.97 (t, 12H, Bu_4_N^+^), 0.88 (t, 6H, CH_3_). ^13^C NMR (DMF-d7, 75.49 MHz) ppm: 59.4 (Bu_4_N^+^), 39.2 (SCH_2_), 32.8 (SCH_2_CH_2_), 30.6, 30.4, 30.3, 30.2, 29.6, 29.1, 28.0 (C3-C10), 24.7 (Bu_4_N^+^), 23.6 (CH_2_CH_3_), 20.6 (Bu_4_N^+^), 14.8 (CH_3_), 14.3 (Bu_4_N^+^). IR (CCl_4_) cm^−1^: 2962, 2926, 2874, 2854, 1469, 1418, 1379, 1308, 1242, 1150, 1121, 968, 895, 839, 787, 759, 432.

**(*n*-Bu_4_N)[2-B_10_Br_9_S(*n*-C_18_H_37_)_2_] (6).** From (*n*-Bu_4_N)[2-B_10_H_9_S(*n*-C_18_H_37_)_2_] (200 mg, 0.22 mmol) and Br_2_ (298 µL, 5.94 mmol), (*n*-Bu_4_N)[2-B_10_Br_9_S(*n*-C_18_H_37_)_2_] (297 mg, 0.18 mmol) was obtained. Yield: 84%. Anal. calc. of C52H110B10Br9NS: C, 38.8; H, 6.9; N, 0.9; S, 2.0; Found: C, 38.7; H, 6.7; N, 0.7; S, 1.7. ^11^B NMR (DMF-d7, 96.32 MHz) ppm: −0.4 (s, 1B), −4.9 (s, 1B), −13.2 (s, 8B). ^1^H NMR (DMF-d7, 300.3 MHz) ppm: 3.60 (m, 4H, SCH_2_), 3.39 (m, 8H, Bu_4_N^+^), 1.95 (m, 2H, SCH_2_CH_2(a)_), 1.88 (m, 2H, SCH_2_CH_2(b)_), 1.77 (m, 8H, Bu_4_N^+^), 1.39 (m, 8H, Bu_4_N^+^), 1.29 (m, 18H, C3-C11), 0.97 (t, 12H, Bu_4_N^+^), 0.88 (t, 6H, CH_3_). ^13^C NMR (DMF-d7, 75.49 MHz) ppm: 59.4 (Bu_4_N^+^), 39.2 (SCH_2_), 32.9 (SCH_2_CH_2_), 30.6, 30.5, 30.3, 30.2, 29.6, 29.0, 28.0 (C3-C16), 24.6 (Bu_4_N^+^), 23.6 (CH_2_CH_3_), 20.6 (Bu_4_N^+^), 14.8 (CH_3_), 14.2 (Bu_4_N^+^). IR (CCl_4_) cm^−1^: 2961, 2925, 2874, 2854, 1468, 1406, 1378, 1308, 1243, 1150, 1121, 968, 895, 838, 786, 762, 433.

## 5. Conclusions

Perbrominated sulfonium derivatives of the *closo*-decaborate anion with alkyl substituents of various hydrocarbon chain lengths can be easily prepared with elemental bromine from their non-halogenated compounds in about a day. The melting points of the compounds obtained are practically the same as their perchlorinated analogues. It can be argued that they depend little on the type of halogen atom, but significantly decrease with the growth of the chain of the alkyl substituent. According to quantum chemical calculations and Hirschfeld surface analysis, the crystal structures are stabilised mainly by cation–anion and anion–anion interactions. The nature of Br…Br interactions between brominated borane fragments was found to be attractive, in contrast to intramolecular Br…Br interactions between adjacent bromine atoms.

## Data Availability

All spectra and XRD data are available from the authors.

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
