# Peer review of "Non-Covalent Interactions in the Crystal Structures of Perbrominated Sulfonium Derivatives of the closo-Decaborate Anion"

_ijms, 2022, doi:10.3390/ijms231912022_

Round 1
Reviewer 1 Report
The manuscript by Golubev et al. describes five crystal structures of salts containing the tetrabutylammonium cation and five different derivatives of perbrominated closo-decaborate anion. The analysis of crystal packing and intermolecular weak interactions takes advantage also of the Hirshfeld’s surface method and of the QTAIM theory and the NCI method using quantum chemical calculations. Overall this work is well performed and generally well described. Nevertheless, no explanations are given about why did authors choose these very particular salts and why studying the weak interactions in such kind of systems should be particularly relevant for the scientific community. Lacking of these argumentations the paper cannot be accepted for publication.
Other points are as follows:
- Can the Authors explain how is possible that the deca-brominated derivatives and their chloridrated analogoues have similar melting points?
- The descriptions and figures for compounds 5 and 6 are not clear enough and the overall crystal packings are not easily understood
- Section 3 “Discussion” is empty.
Author Response
Reviewer 1
Comment
The manuscript by Golubev et al. describes five crystal structures of salts containing the tetrabutylammonium cation and five different derivatives of perbrominated closo-decaborate anion. The analysis of crystal packing and intermolecular weak interactions takes advantage also of the Hirshfeld’s surface method and of the QTAIM theory and the NCI method using quantum chemical calculations. Overall this work is well performed and generally well described. Nevertheless, no explanations are given about why did authors choose these very particular salts and why studying the weak interactions in such kind of systems should be particularly relevant for the scientific community. Lacking of these argumentations the paper cannot be accepted for publication.
Responce
Thank you very much for your comments. We tried to take them into account and made the appropriate corrections.
Perhalogenated derivatives of boron cluster anions are currently being studied in the aspect of their application as ionic liquids or ionic crystals, where weak interactions play a decisive role. However, there are very few theoretical studies in this area for these anions. Our work is intended to bring some clarity into the nature of X…X, X…H interactions for such compounds.
Comment
Other points are as follows:
- Can the Authors explain how is possible that the deca-brominated derivatives and their chloridrated analogoues have similar melting points?
Response
Extensive research on this issue has not yet been carried out, however, judging by the available data, the most noticeable effect between perchlorinated and perbrominated anions will be in salts with small or no alkyl substituents [https://doi.org/10.1021/ja0007511; https://doi.org/10.1039/c5dt01633a]. However, with an increase in the length of the alkyl chain in a substituent, it is the interactions between alkyl substituents that contribute more and more to the energy of intermolecular interactions. Thus, for salts [C6mim]2[B12Cl11OPr] and [C6mim]2[B12Br11OPr], the melting points are 96 and 161°C, while for salts [C6mim]2[B12Cl11O(C12H25)] and [C6mim]2[B12Br11O(C12H25)] the melting points are 144 and 132°C, respectively.
Comment
- The descriptions and figures for compounds 5 and 6 are not clear enough and the overall crystal packings are not easily understood
Responce
Correct
Comment
- Section 3 “Discussion” is empty.
Responce
Sections have been corrected.
Reviewer 2 Report
The work under review is a qualitative example of research that sheds light on the fundamental aspects of weak interactions that form crystals of decaborate anion derivatives, interesting for elongated organic substituents. Here, a series of six compounds containing the prebromodecaborate sulfonium anion derivatives was prepared and characterized. For most of them, the crystal structures were established by means of single-crystal XRD (scXRD), and a detailed analysis of the obtained data was carried out. The experimental results were supported by quantum chemical simulations (QTAIM theory and NCI method). In general, the presented study undoubtedly corresponds to the theme of the Crystals. However, a number of important points need to be clarified.
1) The practical direction of the work should be more clearly clarified. In the Introduction, the authors stated "the creation of new types of materials with desired physicochemical properties" (lines 41-42). Concretize, please, what properties are meant in this work, and how they are related to the crystal structure of the objects of the current study. In addition, to outline the scientific field, it should be indicated which derivatives containing perbrominated derivatives of the closo-decaborate anion have previously been obtained and characterized.
2) It would be useful to discuss in more detail the molar ratio of reactants sufficient for complete bromination of the anion.
3) It is expected that the found features of the crystal structure and especially intermolecular interactions can be reflected in the increase/decrease in the melting temperature. However, this point is not discussed in the text.
4) The authors point to the closeness of the spectra and melting points of the compounds studied here and perchlorinated counterparts. However, further comparison of the structure of the compounds and percolated analogs becomes minimal. Please expand it. It is especially interesting how the types (attractive/repulsive) and the energy of weak interactions in crystals will change when chlorine is replaced by bromine.
5) It will be useful to add powder XRD data to prove the identity of the scXRD data and the polycrystalline samples obtained as a result of the synthesis.
6) For a correct comparison, please indicate the temperature of the calculations in comparison with the experimental scXRD data.
Technical points:
1) The template duplication (Part 3, Discussion) should be removed.
2) Table 1 requires the addition of error values.
3) Taking into account the accuracy of standard methods of elemental analysis, the obtained values should be rounded off. For example, "S, 2.5" would be more correct than "S, 2.48".
Author Response
Rewiever 2
Comment
The work under review is a qualitative example of research that sheds light on the fundamental aspects of weak interactions that form crystals of decaborate anion derivatives, interesting for elongated organic substituents. Here, a series of six compounds containing the prebromodecaborate sulfonium anion derivatives was prepared and characterized. For most of them, the crystal structures were established by means of single-crystal XRD (scXRD), and a detailed analysis of the obtained data was carried out. The experimental results were supported by quantum chemical simulations (QTAIM theory and NCI method). In general, the presented study undoubtedly corresponds to the theme of the Crystals. However, a number of important points need to be clarified.
Response
Thank you very much for your comments. We tried to take them into account and made the appropriate corrections.
Comment
1) The practical direction of the work should be more clearly clarified. In the Introduction, the authors stated "the creation of new types of materials with desired physicochemical properties" (lines 41-42). Concretize, please, what properties are meant in this work, and how they are related to the crystal structure of the objects of the current study. In addition, to outline the scientific field, it should be indicated which derivatives containing perbrominated derivatives of the closo-decaborate anion have previously been obtained and characterized.
Response
Perhalogenated derivatives of boron cluster anions are currently being studied in the aspect of their application as ionic liquids or ionic crystals, where weak interactions play a decisive role. However, there are very few theoretical studies in this area for these anions. Our work is intended to bring some clarity into the nature of X…X, X…H interactions for such compounds.
Comment
2) It would be useful to discuss in more detail the molar ratio of reactants sufficient for complete bromination of the anion.
Response
To completely replace all 9 hydrogen atoms in the boron cage with bromine atoms, 9 elemental bromine molecules are required. Its 3-fold excess is necessary for complete halogenation of the derivative within a day. If a smaller amount of elemental bromine is used, a mixture of fully and partially halogenated substituted derivatives is observed in the reaction mixture over the same time period. With an elemental bromine/cluster ratio of 2/1, the halogenation process can also be completed, but it is necessary to increase the reaction time to 3 days or more.
Information added to section 2. Results. Synthesis.
Comment
3) It is expected that the found features of the crystal structure and especially intermolecular interactions can be reflected in the increase/decrease in the melting temperature. However, this point is not discussed in the text.
Response
We have only four salt in our sampling to establish the reliable correlations. On the first glance, the values of a number of anion properties (surface area, volume, total energy of cation-anion and anion-anion interactions) are inversely proportional to corresponding melting points.
Most probably the value of melting point in salts studied is defined by molecular flexibility of alkyl substituents in anions. From thermodynamic view this contribution can be described as conformational entropy.
Comment
4) The authors point to the closeness of the spectra and melting points of the compounds studied here and perchlorinated counterparts. However, further comparison of the structure of the compounds and percolated analogs becomes minimal. Please expand it. It is especially interesting how the types (attractive/repulsive) and the energy of weak interactions in crystals will change when chlorine is replaced by bromine.
Response
Previously, the structure and intermolecular interactions were studied for several salts of with B10Cl10 anions and organic Et3NH+, Ph4P+, and [Ag(NH3)2]+ cations using QTAIM theory [https://doi.org/10.1016/j.ica.2016.03.025]. The Cl...Cl distances in general up to 0.2 angstroms shorter than Br...Br ones. The energy of Cl...H and Cl...Cl interactions are almost the same as those for Br...H and Br...Br. Thus, the closeness of the spectra and melting points can be easily explained by similarity of the energies of cation-anion or anion-anion interactions. Unfortunately, in this article the NCI analysis was not used to reveal the attractive/repulsive of CL...Cl and Cl...H interactions, however, in our opnion, the situation is the same as in the case of the salts studied in present paper.
Comment
5) It will be useful to add powder XRD data to prove the identity of the scXRD data and the polycrystalline samples obtained as a result of the synthesis.
Response
Done. Powder XRD data added to SI_part 1.
Comment
6) For a correct comparison, please indicate the temperature of the calculations in comparison with the experimental scXRD data.
Response
Added, see section 3. Materials and Methods. Computational details.
All quantum chemical calculations were performed at 0K.
Comment
Technical points:
1) The template duplication (Part 3, Discussion) should be removed.
Response
Done.
Comment
2) Table 1 requires the addition of error values.
Response
Correct.
Comment
3) Taking into account the accuracy of standard methods of elemental analysis, the obtained values should be rounded off. For example, "S, 2.5" would be more correct than "S, 2.48".
Response
Correct.
Round 2
Reviewer 1 Report
Section 3 “Discussion” has not been corrected. In the revised manuscript version, as well in the first version, the section 3 is reported as in the template file downloadable from the journal site
3. Discussion
Authors should discuss the results and how they can be interpreted from the per-spective of previous studies and of the working hypotheses. The findings and their im-plications should be discussed in the broadest context possible. Future research direc-tions may also be highlighted.
This point should be changed
Author Response
Thank you very much for your work on improving our article.
We have corrected Section 3 "Discussion". It presents a discussion of the results of the work and how they can be interpreted with previously written articles.
Reviewer 2 Report
The authors responded in detail to the comments and sufficiently modified the manuscript.
Author Response
Thank you very much for your work on improving our article.